# E-nose, E-tongue Combined with GC-IMS to Analyze the Influence of Key Additives during Processing on the Flavor of Infant Formula

**DOI:** 10.3390/foods11223708

**Published:** 2022-11-18

**Authors:** Xuelu Chi, Hongxia Guo, Yangdong Zhang, Nan Zheng, Huimin Liu, Jiaqi Wang

**Affiliations:** 1College of Animal Science, Xinjiang Agriculture University, Urumchi 830091, China; 2Key Laboratory of Quality & Safety Control for Milk and Dairy Products of Ministry of Agriculture and Rural Affairs, Institute of Animal Sciences, Chinese Academy of Agricultural Sciences, Beijing 100193, China; 3State Key Laboratory of Animal Nutrition, Institute of Animal Sciences, Chinese Academy of Agricultural Sciences, Beijing 100193, China

**Keywords:** infant formula, headspace-gas chromatography-ion mobility spectrometry, volatile compounds, E-tongue, E-nose

## Abstract

In order to analyze the influence of key additives during processing on the flavor of infant formula, the headspace-gas chromatography-ion mobility spectrometry, electronic tongue, and electronic nose techniques were used to evaluate flavor during the processing of stage 1 infant formula milk powder (0–6 months), including the analysis of seven critical additives. A total of 41 volatile compounds were identified, involving 12 aldehydes, 11 ketones, 9 esters, 4 olefins, 2 alcohols, 2 furans, and 1 acid. The electronic nose metal oxide sensor W5S had the highest response, followed by W1S and W2S, illustrating that these three sensors had great effects on distinguishing samples. The response results of the electronic tongue showed that the three sensory attributes of bitter, salty, and umami, as well as the richness of aftertaste, were more prominent, which contributed significantly to evaluating the taste profile and distinguishing among samples. Raw milk is an essential control point in the flavor formation process of stage 1 infant formula milk powder. Demineralized whey powder is the primary source of potential off-flavor components in hydrolyzed milk protein infant formula. This study revealed the quality characteristics and flavor differences of key additives in the production process of stage 1 infant formula milk powder, which could provide theoretical guidance for the quality control and sensory improvement of the industrialized production of infant formula.

## 1. Introduction

Flavor is one of the main drivers for consumer food preference [1]. In dairy products, flavor chemistry is used to uncover the process of flavor perception by finding specific compounds responsible for specific odors, tastes, or chemical aesthetics [2,3]. Infancy is the first critical period for growth and development. The taste development in early infants is closely related to the food composition during pregnancy and lactation, which also affects the food exposure degree and the ability of accepting food during childhood [4,5,6]. In addition, the effect of taste experience on later physiology and behavior in early postnatal development has been verified by animal experiments [7]. The taste stimulation for infants, which is brought by amniotic fluid, breast milk, and formula powder, plays a vital part in the subsequent feeding degree and dietary preferences of infants. Breast milk has a comprehensive nutritional composition and is the most ideal food for newborn babies. However, due to the insufficient breast milk of many mothers, stage 1 infant formula milk powder is one of the food sources for infants with incomplete development of digestion and absorption functions. Therefore, it is particularly important to study the flavor substances of infant formula milk powder.

In the process of evaluating flavor at the molecular level, gas chromatography–mass spectrometry (GC-MS) is frequently used for the identification of volatile organic compounds in dairy. Compared with this traditional method, headspace-gas chromatography-ion mobility spectrometry (HS-GC-IMS), which is an emerging technology, has many advantages, such as low detection limit, fast pretreatment, and intuitive analysis [8,9]. It also enables the visual analysis of volatile components, establishment of fingerprints, and separation of isomers [10,11]. Electronic tongue (E-tongue) and electronic nose (E-nose) can simulate the olfactory system and gustatory perceptions of humans, both consisting of an array of sensors and suitable pattern recognition [12,13]. They are often used in the food industry for food assessment and to differentiate the sensory quality by describing the overall odors and the nuances in taste [11]. The application of intelligent sensory analysis compensates for the individual differences among human subjects and expands the application scope of descriptive sensory analysis. For samples that cannot be tasted directly, analytical conditions are available.

Research on the flavor of infant formula milk powder has mainly focused on the lipid oxidation [14,15], the hydrolysis of milk protein [16,17,18,19,20], different breeds or raw materials [21], and the difference between infant formula milk powder and breast milk [22,23]. However, factors that may affect the flavors of milk powder products are speculated. Stage 1 infant formula is supplement nutrition for newborn babies aged 0–6 months. Compared with the infant formulas in other stages, the added supplements and the aroma components are simpler. By analyzing the basic products, not only can the importance of raw materials be better understood, basic data support for subsequent research is also provided.

In this study, we obtained all the added ingredients in the production of stage 1 infant formula and discussed the sensory quality differences between key additives through GC-IMS combined with intelligent sensory technology (electronic nose and electronic tongue). The purposes of this study were to investigate the evolution of flavor during processing, speculate the factors that may affect the sensory quality of infant formula milk powder products, which might provide reference for flavor quality control in the production process, and present new insights into improving milk powder aroma.

## 2. Materials and Methods

### 2.1. Samples

Samples from various milk powder processing stages were obtained for analysis at a large dairy company in China. A total of seven key original supplementary materials were collected from the stage 1 infant formula production process. There was raw milk (RM), pasteurized milk (PM), demineralized whey (DW), demineralized whey powder (DWP), hydrolyzed whey protein powder (HWP), whey protein powder supplemented with alpha lactalbumin (WPA), and infant formula (IF). All the samples were stored at −20 °C until testing.

### 2.2. GC–IMS Analysis

Analyses of flavor components were performed on a GC-IMS Flavor Spec (G.A.S, Beijing, China). The chromatographic column used for the analyses was a non-polar capillary column MXT-5 (15 m × 0.53 mm × 1.0 μm). Samples of 3 mL were placed in 20 mL headspace vials. The samples were incubated at 40 °C for 30 min and rotated at 500 rpm during incubation.

The column temperature was kept at 60 °C. Nitrogen (99.999%) was used as the carrier gas. The flow rate was first set at 2 mL/min for 2 min, then increased to 10 mL/min within 8 min, then increased to 100 mL/min within 10 min, and finally increased to 150 mL/min within 10 min and held for 5 min.

### 2.3. E-nose Analysis

Volatile flavor was analyzed using a PEN 3 electronic nose (Airsense Analytics Inc., Schwerin, Germany) with ten metal oxide conductivity gas sensors. They consisted of a sample acquisition system, a metal oxide sensor array, a data acquisition system, and a Win Muster signal processing software. The main groups were detected, and the corresponding ranges of sensitive gases are shown in Table 1.

The 8.0 mL sample was placed into a 20 mL headspace vial. The incubation temperature was 40 °C ± 2 °C, and equilibration was carried out at a speed of 960 rpm/min for 300 s. The E–nose was applied with a detection time of 200 s, a cleaning time of 300 s, and an injection flow rate of 300 mL/min. After each sample analysis, the system was purged with filtered air for 300 s before the next sample injection to reestablish the instrument baseline. To ensure the accuracy of the E-nose test results, three groups were performed in parallel, and stable data were selected for statistical analysis during the measurement process.

### 2.4. E-tongue Analysis

E-tongue analysis was operated using the Taste-Sensing System SA 402B (Intelligent Sensor Technology Co., Ltd., Atsugi, Japan). The E-tongue device was equipped with 5 sensors (CT0, CA0, C00, AE1, and AAE) that were sensitive to the sour, salty, umami, bitter, and astringent tastes of the sample (Table 2). They also rated bitter aftertaste, astringent aftertaste, and richness.

The samples were first thawed at room temperature, properly diluted, and filtered through double-layer gauze. The taste sensors and reference electrodes of the E-tongue were pre-activated for more than 24 h. A 30 mL sample was added into a special sample cup in duplicate. The sample was measured repeatedly more than 4 times. After filtering and correcting the data, E-tongues turned electrical signals into relish signals to reflect the taste information of the sample through built-in plug-ins. They calculated the theoretical charge density at the membrane surface using the Gouy–Chapman theory and the Poisson–Boltzmann equation and then investigated the lipid/polymer membrane’s responses to sour, umami, salty, bitter, astringent, richness, bitter aftertaste, and astringent aftertaste.

### 2.5. Statistical Analysis

Volatile substance data was performed using the LAV software (version 2.2.1), GC-IMS Library Search, Reporter plugin, Gallery plot plugin, and Dynamic PCA plugin in GC-IMS. The odor profile and taste attribute scores of the samples were calculated using excel. The taste profile and principal component analysis of the samples were analyzed using the plugin of the E-tongue. Correlation analysis was performed using The Unscrambler X×10.4. Thermographic analysis was drawn through Prism.

## 3. Results and Discussion

### 3.1. Volatile Organic Compounds Result

The volatile components in the original supplementary material samples at whole processing stages were carried out by GC-IMS. The retention time and migration time were combined for qualitative analysis. The 3D topographic map can intuitively express the difference (Figure 1A). The volatile organic compound types in the critical additives were slightly similar, but the signal intensities were quite different. The volatile organic components of the 2D topographic plots of the critical additions at whole processing stages are shown in Figure 1B. The background of the GC-IMS spectra is blue, and the red vertical line at abscissa 1.0 is the reactive ion peak (RIP) after normalization. In the topographic plot, the ordinate represents the retention time (s) of the gas chromatogram, and the abscissa represents the ion migration time. The colors indicate the signal strength of the compounds, with the white color as the lower concentration and the red color as the higher concentration; the intensity increases as the color deepens. After the data was normalized, each point on the right side of the reactive ion peak marked a volatile flavor substance or its dimer [25,26].

The Figure 1 shows that most of the differences are reflected in the first 300 s of the analysis. The main components of the difference are small molecules and low boiling-point compounds, which can quickly separate and analyze the peaks in the chromatographic column. Figure 1B compares demineralized whey powder, hydrolyzed whey protein powder, and whey protein powder supplemented with alpha lactalbumin samples; the amount and color of speckles in raw milk, pasteurized milk, and demineralized whey samples were more significantly intense, which indicated that both the quantity and content of volatile constituents were higher. Raw milk sample was used as a guide in Figure 1C, and the topographic plot of the other six critical additions were subtracted from the base value. The background is shown in white after subtracting the same components. However, red represents the higher concentration of the volatile component, and lower values are indicated by blue [27,28].

The volatile compounds in the seven key original supplementary materials were qualitatively performed using the spectral library with GC-IMS. The identities of the compounds were determined using double comparisons of retention time and migration time (Table 3). There were altogether 41 flavor compounds that were identified, including the monomer and its polymer, 12 aldehydes, 2 alcohols, 11 ketones, 9 esters, 4 olefins, 1 acid, and 2 furans. It was detected that one analyte might produce multiple signals, which included protonated monomers and proton-bound dimers or trimers [28]. IMS provided the second separation of compounds that was closely related to the high proton affinity or concentration of the compounds in the analytes [27,28]. Some high concentration components were performed by both monomer and dimer forms, such as 2-heptanone, ethyl hexanoate, heptane, and beta-Pinene.

To clarify the variances in the compositions of volatile compounds, all peaks were selected for fingerprint comparisons (Figure 2). Each row in the fingerprint spectrum represents a signal peak in the total chosen for analysis, and each column represents the signal response of the same volatile substances in the samples.

The substances in the A region could be used as the characteristic of raw milk for its higher content. There were ethyl caproate, ethyl butyrate, ethyl caprylate, butyl acetate, 2,3-butanedione, ethanol, ethyl propionate, ethyl pentanoate, and acetone. Esters can be synthesized by alcoholysis or chemical esterification of fatty acids and alcohols [29]. For example, it has been reported that ethyl butyrate was obtained by esterification of butyric acid with ethanol. Esters are unique flavor components in raw milk, usually described as sweet and fruity. It is considered a flavor description that positively contributes to the sensory quality and weakens the pungent taste of fatty acids and the bitter taste of amino groups [30,31,32].

There was a high imprint similarity between raw milk and pasteurized milk. The substances in the B region were the unique components with high contents in pasteurized milk. These differential components were mainly caused by thermal processing. As a result of the limitations of the GC-IMS spectral library, the substances in the B region were difficult to qualitatively analyze. It was reported that these compounds may be aldehydes, ketones, or sulfur-containing compounds, such as carbon disulfide (CS_2_), hydrogen sulfide (H_2_S), and dimethyl sulfide (DMS), etc. They act as aroma components and precursors in reactions, producing more complex aroma compounds [33,34,35].

The volatile compounds, which were detected in demineralized whey, included a variety of components, mainly phenylacetaldehyde, β-pinene, methyl heptanoate, pentanol, isobutyraldehyde, isobutyl acetate, 4-methyl-2-pentanone, 2-acetyl furan, 3-methyl butyric acid, and 3-methyl butyraldehyde. They had the strongest specificity and played an important role in distinguishing demineralized whey from other samples, including a variety of components, mainly phenylacetaldehyde and β-pinene. Aldehydes are typically formed from amino acids, converted into intermediate imines by enzymatic transacylation, and then decarboxylated or degraded by Strecker [35,36]. These compounds usually have lower thresholds and more significant impacts on the overall flavor.

The substances marked in the C region were unique components with high contents in hydrolyzed whey protein powder, including 2-octanone, 2-heptanone, and α-phellandrene. Breast milk and infant formula are rich in lipid-derived volatile compounds, consisting mainly of carbonyl and alcohol compounds. Infant formula contains more heat-treatment-related volatiles than breast milk [14,16], and terpenes are common in formula and breast milk [17,22]. Carbonyl compounds, such as 2-ketones and aldehydes, are thought to be the crucial aromatic components in milk. They are often described as specific aromas even at lower thresholds, such as fishy, rancid, paint-like, soapy, metallic, green, and fruity [18]. When the content exceeds a certain value, it will have an oxidation and spoilage odor, with negative impacts on the flavor. The compounds marked in area D are used to distinguish the hydrolyzed whey protein powder sample from the others. However, these compounds are not accurately qualitative in the GC-IM spectrum library. Therefore, we only marked signals in the spectrum but gave no accurate material information.

As shown in the E region, the contents of (E)-2-decenal, ethyl 3-methylbutyrate, and valeraldehyde in whey protein powder supplemented with alpha lactalbumin were higher than those of the other sample groups. It has been reported that (E)-2-Decenal contributes to the fat taste in infant formula but produces an unpleasant jarring taste when present at high levels, which can be used as an indicator of product spoilage [37,38].

The highest levels in infant formular were trans-2-heptenal and benzaldehyde, marked in the F region. Carbonyl compounds and alcohols have been reported to be the key components in breast milk and infant formula. Benzaldehyde has been reported as a key aromatic active compound, described as an almond flavor [23]. (E)-2-heptenal was also detected in infant formula and was described as a fruity and soapy odor that contributed significantly to the odor of the milk powder [39]. Usually, the content in infant formula milk powder is higher than breast milk [23]. Most of the production of flavor substances was related to the degradation and metabolism of flavor precursors, such as amino acids and fatty acids.

To highlight the differences among seven key original supplementary materials, PCA and FSA were indicated based on the signal intensity of the compounds (Figure 3). The contribution rates of PC1 and PC2 were 50% and 21%, and the cumulative contribution rate was 71% (>70%). The PCA diagram showed a clear separation between each group of seven crucial additions and did not overlap. It can be seen from Figure 3 that except for some outliers, the score distribution is relatively compact, indicating that all samples have consistent responses to the system. The loading plot indicated that the radial length of each variable was obviously different, which shows that these compounds have different influences on distinguishing samples. The responses of all the monomers and their polymers were selected as variables, and their positions in the figure were consistent with the results of the fingerprint spectra. The whey protein powder supplemented with alpha lactalbumin was close to demineralized whey powder, hydrolyzed whey protein powder, and infant formula. It indicated that the detected volatile organic compounds were relatively similar. There were obvious distance differences between these four samples and the demineralized whey, pasteurized milk, and raw milk.

As shown in Figure 4, the bottom area showed the normal distribution for each sample. The longer distance indicated a larger pronounced difference. However, the volatile organic compounds in demineralized whey powder, hydrolyzed whey protein powder, and whey protein powder supplemented with alpha-lactalbumin were similar. The FSA results were basically the same as those of the principal component analysis; therefore, it further confirmed the conclusions.

### 3.2. E-nose Analysis Results

In Figure 5, the W5S, W1S, and W2S sensors had intense responses to the odors of seven key additions, and the differences in flavor were mainly distinguished by W5S, W1S, and W2S. Compared with these sensors, the responses of the W3C, W1C, and W5C sensors were slightly weaker. The response differences between the samples were easy to distinguish. There were significant differences in the contents of aromatic compounds, benzenes, amino olefins, exercise aromatics, and long-chain alkanes. The changes of the W2W and W6S sensors were not significant, and the contents of sulfur- and chlorine-containing compounds were lower; there were no significant differences, which did not contribute much to the discrimination of the samples.

Partial least square regression analysis (PLSR) was used to simplify the data structure and carry out correlation analysis between the two groups of variables, which further explained the potential relationship between the electronic nose sensors and the compounds identified by GC-IMS. The correlation load diagram is shown in Figure 6 with 10 sensors of the E-nose as the X variable and the response of the VOCs as the *Y* variable. There were many compounds detected by GC-IMS, but their contributions to the overall flavor varied greatly due to their different threshold values and contents. Compounds indicated that they can be well-explained by the model located only between the two ellipses. Most of the compounds had a high correlation with the E-nose sensors, especially esters. We believe that these VOCs contributed a lot in distinguishing samples.

### 3.3. E-tongue Analysis Results

As shown in Table 4, raw milk and pasteurized milk had lower responses to astringency aftertaste; other samples had no response to the two taste attributes of sourness and astringency. In terms of bitterness attributes, all samples had higher responses, and demineralized whey powder samples had the most prominent response, followed by hydrolyzed whey protein powder and infant formula, which was consistent with previous studies. Hydrolyzed protein refers to the fragmentation of complete proteins into small molecular proteins, peptide fragments, or amino acids, which are obtained by enzymatic hydrolysis or acid hydrolysis to reduce the sensitization of macromolecular proteins [19]. Infants fed hydrolyzed protein formula often suffer from indigestion, which can be somewhat relieved by the use of hydrolyzed IF with a flavorful taste. Bitterness is strongly related to the degree of hydrolysis and may also be affected by enzymes used for hydrolysis and the hydrolysis process. Limited bitterness in foods is acceptable, but high levels of bitter peptides and bitter amino acids can limit consumer choice [20,22,40]. At present, there is little research on the flavor of hydrolyzed protein milk powder, and the research of hydrolyzed protein infant formula milk powder and complementary food is also a topic worthy of further study.

The salty properties of demineralized whey and demineralized whey powder were significantly lower than those of other components, which was consistent with their own processes [15,16]. Hydrolyzed whey protein powder was significantly different from other additions in the taste evaluation, and the response to the salty taste sensor was the strongest, followed by the umami taste sensor. The raw milk and pasteurized milk also scored higher on the savory attribute [41,42]. There were no differences in the performances of astringent and bitter aftertastes for any of the samples. The responses of the samples to the sour taste were different, but they were all below the tasteless point, which was not within the perception range. Hydrolyzed whey protein powder had the highest response in terms of richness, and all samples responded with significant differences.

The signal responses of the five artificial lipid membrane sensors were analyzed in pairs using the analysis software of the E-tongue instrument. As shown in the scatter plot of saltiness–astringency and saltiness–bitterness (Figure 7A), hydrolyzed whey protein powder, demineralized whey powder, and infant formula indicated obvious outliers, which may be due to the fact that these three samples underwent the spray-drying process and were more consistent in taste. The outlier status of demineralized whey may be due to the special nature of the desalted whey, which is desalinated and therefore, has a lower response to the salty taste attribute. Pasteurized milk, raw milk, and alpha protein-added whey protein powder were closely distributed in the astringency–salty group distinction. Hydrolyzed whey protein powder showed a significant difference with other samples in the process of salty group distinction (Figure 7B).

Most of the sensors showed a negative correlation in E-nose and E-tongue (Figure 8). Sourness and Aftertaste-A showed positive correlations with ten sensors of E-nose, respectively. There was a highly positive correlation between Aftertaste-A and W3S. Bitterness, astringency, and umami showed highly negative correlations with the ten sensors of E-nose except W1W. In this experiment, the samples were distinguished by evaluating the volatile flavors and tastes. The functions of E-nose and E-tongue are related to a certain extent. These two methods were combined to evaluate the samples, which guaranteed the accuracy of the results.

## 4. Conclusions

In this study, the diversity of flavor compounds in critical additions at the first stages of the infant formula processing line were analyzed by HS-GC-IMS, E-tongue, and E-nose. In total, 41 volatile substances were identified, including the monomer and its polymer, by HS-GC-IMS in seven key original supplementary materials. There were 12 aldehydes, 2 alcohols, 11 ketones, 9 esters, 4 olefins, 1 acid, and 2 furans. The intelligent sensory technology could distinguish between the differences in flavor of each critical addition.

The contents of characteristic volatile compounds in raw milk were much higher than in the other samples. Raw milk is an essential control point in the flavor formation process of stage 1 infant formula milk powder. The quality of raw materials determines the quality of the final product. Demineralized whey powder is the primary source of potential off-flavor components in hydrolyzed milk protein infant formula. Through the rational application of HS-GC-IMS, combined with intelligent sensory technology, the in-depth understanding of the flavor formation of infant formula in the production line can provide potential ideas for improving processing methods, quality control of raw milk, and optionally added excipients.

## Figures and Tables

**Figure 1 foods-11-03708-f001:**
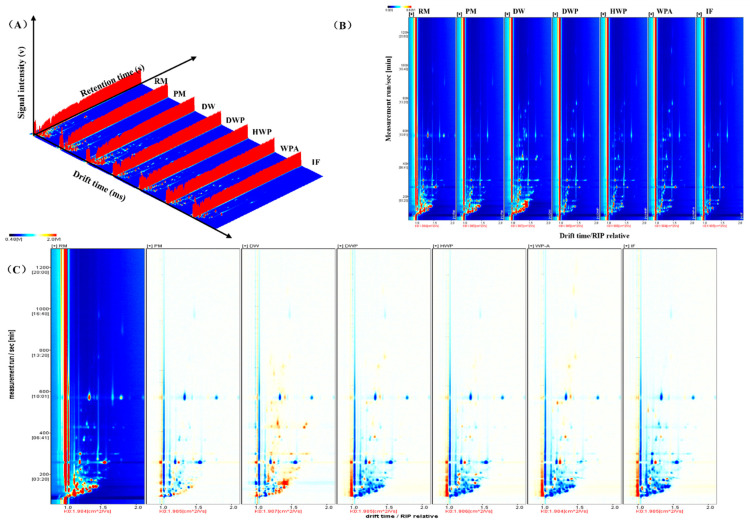
GC-IMS analysis results: (**A**) 3D-topographic, (**B**) topographic plot of GC-IMS spectra, and the (**C**) comparison results under the spectral diagram. RM, raw milk; PM, pasteurized milk; DW, demineralized whey; DWP, demineralized whey powder; HWP, hydrolyzed whey protein powder; WPA, whey protein powder with added alpha-lactalbumin; IF, infant formula.

**Figure 2 foods-11-03708-f002:**
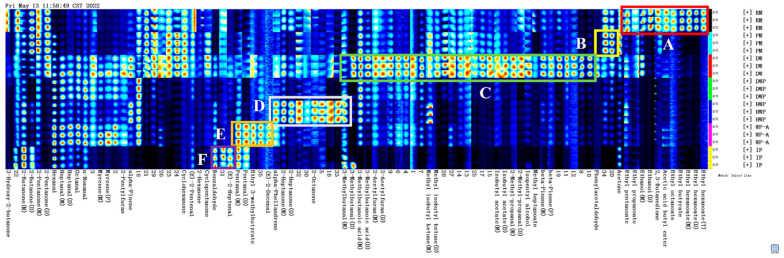
GC-IMS fingerprint spectra. A–F region represent the characteristics of sample RM, PM, DW, DWP, HWP; WPA and IF. RM, raw milk; PM, pasteurized milk; DW, demineralized whey; DWP, demineralized whey powder; HWP, hydrolyzed whey protein powder; WPA, whey protein powder with added alpha-lactalbumin; IF, infant formula.

**Figure 3 foods-11-03708-f003:**
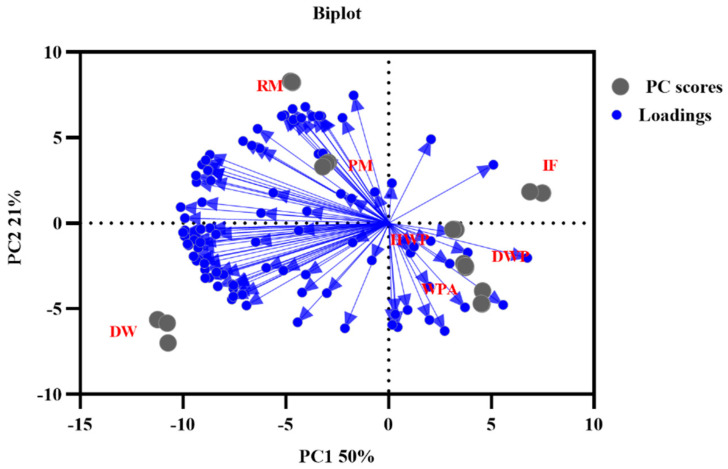
Principal component analysis (PCA) and different variance loading plots according to the quantitative results. RM, raw milk; PM, pasteurized milk; DW, demineralized whey; DWP, demineralized whey powder; HWP, hydrolyzed whey protein powder; WPA, whey protein powder with added alpha-lactalbumin; IF, infant formula.

**Figure 4 foods-11-03708-f004:**
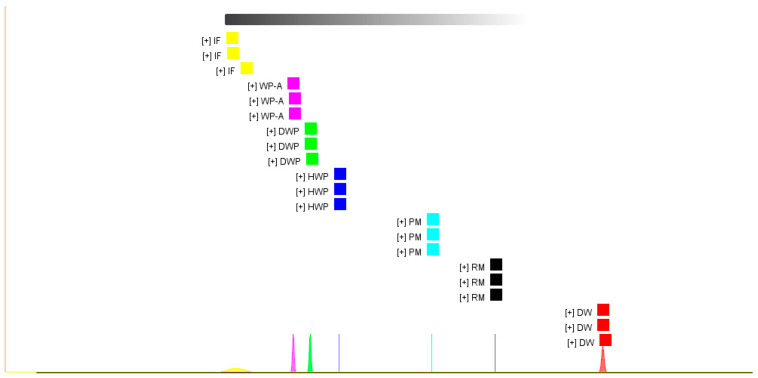
The results of the fingerprint similarity analysis (FSA). RM, raw milk; PM, pasteurized milk; DW, demineralized whey; DWP, demineralized whey powder; HWP, hydrolyzed whey protein powder; WPA, whey protein powder with added alpha-lactalbumin; IF, infant formula.

**Figure 5 foods-11-03708-f005:**
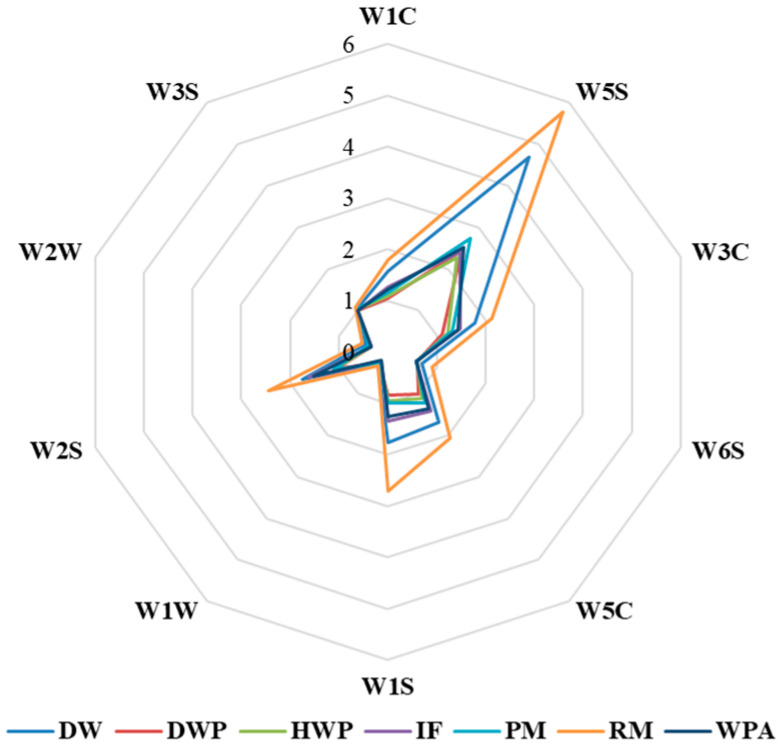
Distribution of the volatile compound species in the samples using E-nose sensors. RM, raw milk; PM, pasteurized milk; DW, demineralized whey; DWP, demineralized whey powder; HWP, hydrolyzed whey protein powder; WPA, whey protein powder with added alpha-lactalbumin; IF, infant formula.

**Figure 6 foods-11-03708-f006:**
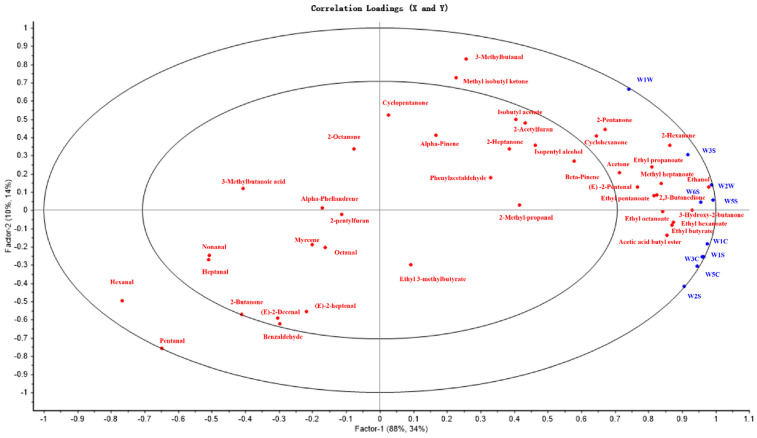
Correlation of sensors and VOCs.

**Figure 7 foods-11-03708-f007:**
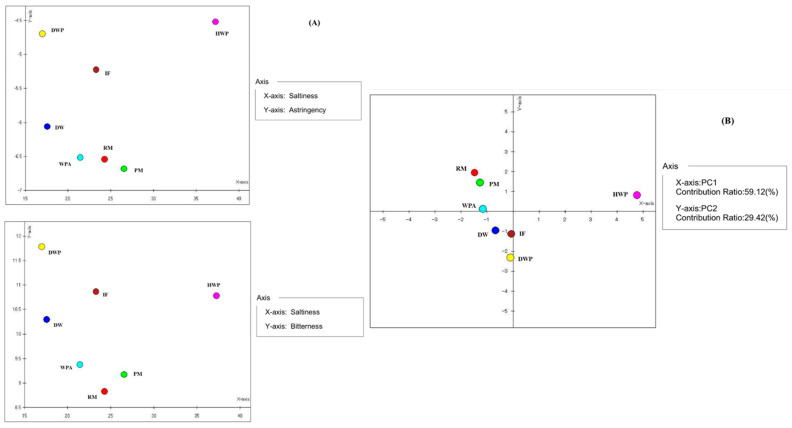
The scatter plot, Saltness-Bitterness and Saltness-Astringency (**A**) and principal component results of E-tongue (**B**). RM, raw milk; PM, pasteurized milk; DW, demineralized whey; DWP, demineralized whey powder; HWP, hydrolyzed whey protein powder; WPA, whey protein powder with added alpha-lactalbumin; IF, infant formula.

**Figure 8 foods-11-03708-f008:**
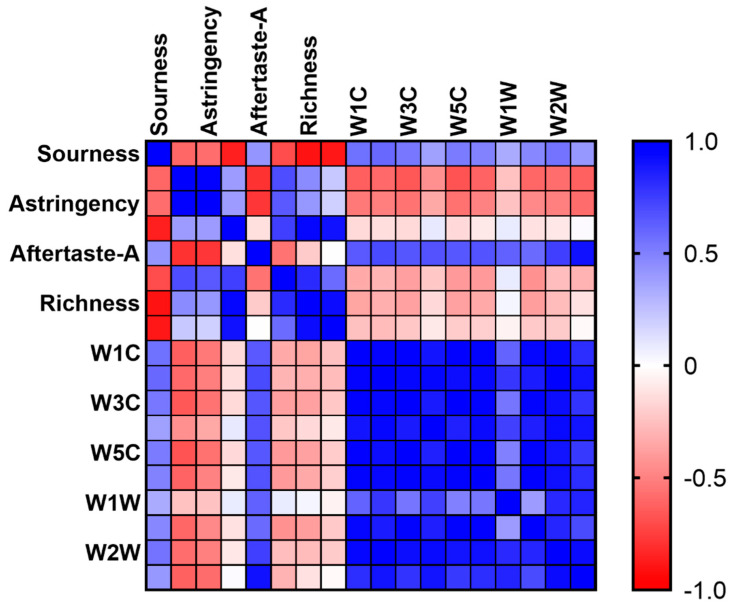
Correlation analysis between E-nose and E-tongue; blue represents a positive correlation, and red represents a negative correlation.

**Table 1 foods-11-03708-t001:** Metal oxide conductivity gas sensors of the E-nose [24].

NO.	Sensor	Performance Description (Sensitivity to)	Main Detection Group	Sensor Corresponding Group Threshold (mL/m^3^)
1	W5S	nitrogen oxides	NO_2_	1
2	W1S	methyl	CH_4_	100
3	W2S	alcohols, ketones, and aldehydes	CO	100
4	W3C	ammonia	C_6_H_6_	10
5	W1C	benzene	C_7_H_8_	10
6	W5C	short-chain aromatic compounds and olefin	C_3_H_8_	1
7	W1W	sulfur compounds	H_2_S	1
8	W2W	organic sulfides	H_2_S	1
9	W6S	hydrogen	H_2_	100
10	W3S	long-chain alkanes	CH_4_	100

**Table 2 foods-11-03708-t002:** Artificial lipid sensors of the E-tongue.

NO.	Sensors	Characteristics
1	CT0	Saltiness
2	CA0	Sourness
3	AAE	Umami, Richness
4	AE1	Astringency, Aftertaste-A
5	C00	Bitterness, Aftertaste-B

**Table 3 foods-11-03708-t003:** GC-IMS compound results.

Count	Compound	CAS#	Formula	MW	RI	Rt [s]	Dt [a.u.]
Aldehyde							
1	2-Methyl-propanal	C78842	C_4_H_8_O	72.1	565.7	132.631	1.10393
2	3-Methylbutanal	C590863	C_5_H_10_O	86.1	630	156.409	1.17527
3	Pentanal	C110623	C_5_H_10_O	86.1	680.7	178.104	1.18418
4	2-Pentenal(E)	C1576870	C_5_H_8_O	84.1	757.2	232.444	1.1137
5	Hexanal	C66251	C_6_H_12_O	100.2	786	257.663	1.25466
6	Heptanal	C111717	C_7_H_14_O	114.2	896.6	381.103	1.33043
7	2-heptenal(E)	C18829555	C_7_H_12_O	112.2	956.9	481.55	1.25549
8	Benzaldehyde	C100527	C_7_H_6_O	106.1	970	506.611	1.15146
9	Octanal	C124130	C_8_H_16_O	128.2	1003.1	572.572	1.41471
10	Phenylacetaldehyde	C122781	C_8_H_8_O	120.2	1063.6	684.933	1.25699
11	n-Nonanal	C124196	C_9_H1_8_O	142.2	1101.3	765.988	1.47438
12	(E)-2-Decenal	C3913813	C_10_H_18_O	154.3	1248.9	1186.104	1.48774
Ketone							
1	Acetone	C67641	C_3_H_6_O	58.1	467.9	103.196	1.12659
2	2,3-Butanedione	C431038	C_4_H_6_O_2_	86.1	554	128.68	1.1711
3	2-Butanone	C78933	C_4_H_8_O	72.1	562.2	131.427	1.06066
4	2-Pentanone	C107879	C_5_H_10_O	86.1	670.9	173.711	1.12572
5	3-Hydroxy-2-butanone	C513860	C_4_H_8_O_2_	88.1	716.1	200.549	1.0539
6	Methyl isobutyl ketone	C108101	C_6_H_12_O	100.2	725	207.06	1.17904
7	2-Hexanone	C591786	C_6_H_12_O	100.2	774.4	247.178	1.19207
8	Cyclopentanone	C120923	C_5_H_8_O	84.1	787.7	259.21	1.10492
9	2-Heptanone	C110430	C_7_H_14_O	114.2	885.7	366.108	1.26143
10	Cyclohexanone	C108941	C_6_H_10_O	98.1	894.2	377.568	1.15368
11	2-Octanone	C111137	C_8_H_16_O	128.2	991	549.617	1.33151
Esters							
1	Ethyl propanoate	C105373	C_5_H_10_O_2_	102.1	696.4	186.891	1.14908
2	Isobutyl acetate	C110190	C_6_H_12_O_2_	116.2	759.4	234.244	1.2371
3	Ethyl butyrate	C105544	C_6_H_12_O_2_	116.2	788.3	259.726	1.20472
4	Acetic acid butyl ester	C123864	C_6_H_12_O_2_	116.2	793.7	264.761	1.62477
5	Ethyl 3-methylbutyrate	C108645	C_7_H_14_O_2_	130.2	855.7	329.38	1.2617
6	Ethyl pentanoate	C539822	C_7_H_14_O_2_	130.2	896.4	380.733	1.2681
7	Ethyl hexanoate	C123660	C_8_H_16_O_2_	144.2	1004	574.147	1.34172
8	Methyl heptanoate	C106730	C_8_H_16_O_2_	144.2	1016.7	596.043	1.35772
9	Ethyl octanoate	C106321	C_10_H_20_O_2_	172.3	1181.1	970.444	1.48244
Olefin							
1	Alpha-Pinene	C80568	C_10_H_16_	136.2	929.8	433.527	1.21834
2	Beta-Pinene	C127913	C_10_H_16_	136.2	971.5	509.514	1.22342
3	Myrcene	C123353	C_10_H_16_	136.2	988.9	545.098	1.21596
4	Alpha-Phellandrene	C99832	C_10_H_16_	136.2	1009	582.713	1.2157
Furan							
1	2-Acetylfuran	C1192627	C_6_H_6_O_2_	110.1	926.8	428.438	1.12462
2	2-pentylfuran	C3777693	C_9_H_14_O	138.2	988.5	544.244	1.25556
Alcohol							
1	Ethanol	C64175	C_2_H_6_O	46.1	437.4	95.431	1.05592
2	Isopentyl alcohol	C123513	C_5_H_12_O	88.1	733.8	213.753	1.24258
Acid							
1	3-Methylbutanoic acid	C503742	C_5_H_10_O_2_	102.1	828.7	299.497	1.20925

Note, MW, molecular weight; RI, retention index; Rt, retention time; Dt, drift time.

**Table 4 foods-11-03708-t004:** The taste values.

CH	Sourness [a.u.]	Saltiness [a.u.]	Bitterness [a.u.]	Astringency [a.u.]	Umami [a.u.]	Aftertaste-A [a.u.]	Aftertaste-B [a.u.]	Richness [a.u.]
Tasteless	−13	−6	0	0	0	0	0	0
RM	−46.87 ^ab^	24.27 ^c^	8.83 ^d^	−6.54 ^d^	14.14 ^e^	0.04 ^a^	0.31 ^b^	1.58 ^bc^
PM	−48.63 ^c^	26.52 ^b^	9.17 ^d^	−6.68 ^d^	14.76 ^d^	0.02 ^b^	0.21 ^c^	2.36 ^b^
DW	−46.2 ^a^	17.55 ^e^	10.30 ^c^	−6.06 ^c^	17.33 ^b^	−0.1 ^c^	0.24 ^c^	0.18 ^d^
DWP	−47.95 ^bc^	16.99 ^e^	11.79 ^a^	−4.70 ^a^	15.55 ^c^	−0.11 ^cd^	0.21 ^c^	0.78 ^d^
HWP	−56.92 ^e^	37.24 ^a^	10.78 ^ab^	−4.52 ^a^	18.99 ^a^	−0.13 ^d^	0.40 ^a^	8.51 ^a^
WP-A	−47.99 ^bc^	21.42 ^d^	9.38 ^d^	−6.52 ^d^	14.86 ^d^	−0.09 ^c^	0.22 ^c^	1.22 ^bcd^
IF	−49.99 ^d^	23.27 ^c^	10.87 ^b^	−5.22 ^b^	15.00 ^cd^	−0.1 ^cd^	0.21 ^c^	0.85 ^cd^
*p*-value	0	0.012	0	0	0.003	0.036	0	0

Note: Different lowercase letters (a–e) in the same row indicate significant differences between different taste values (*p* < 0.05).

## Data Availability

Data are contained within the article.

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
