# Peer review of "E-nose, E-tongue Combined with GC-IMS to Analyze the Influence of Key Additives during Processing on the Flavor of Infant Formula"

_foods, 2022, doi:10.3390/foods11223708_

Round 1
Reviewer 1 Report
This manuscript describes the e-nose, e-tongue and GC_IMS application to analyze the influence of additives on the taste and aroma of infant formula. Although this study seems very interesting especially from an analytical point of view, it nevertheless requires further multivariate statistical processing to combine all the considered variables. Data merged from these techniques can be a powerful tool for achieving greater reliability results and can provide useful complementary information.
For these reason I think that the present article will be the major revision.
Abstract:
line 13-14: e-tongue does not reveal volatile compounds… please change the sentence
line 20: what is the meaning of “richness of aftertaste”? have never heard this definition in relation to e-tongue sensory attributes.
Introduction
Line 31-33: The reference 2 is about off-flavour in meat and meat products, please look for a more suitable reference.
Line 61: different substrates ?????? explain better
Line 64-66: “In this study, we obtained all the added ingredients in the production of infant formula, which combined with intelligent sensory technology, the sensory quality differences between key additives in the milk powder production process”. It is not clear the meaning of this sentence, English needs to be revised.
Materials and methods
Line 77: “whey protein powder hydrolyzed powder (HWP)”- do you mean “hydrolyzed whey protein powder”?
Line 87: delete “first”
Line 88: “Finally, the flow rate was 150 mL/min for 30 min and held for 5 min”… 30 minutes or 5 minutes? it is not clear!
Line 89: “The entire analysis time kept 35 min”- are you sure? According to my calculation the time is: 2+10+20+ (5 or 30) minutes……
Line 101: “equilibration was carried out at a speed of 30 r/min for 5 min”. r???? Please explain the procedure better as it is not clear
Line 106: The sampling time was 200 seconds and the sensor signals were collected only after 6 seconds? Normally after 6 seconds of sampling the sensor signals continue to grow and neither the maximum value nor the plateau have been reached.
Line 111: “salty” is written two times and “umami” is not reported
Line 121: Please define or introduce a reference that explains how the electric signals were converted into taste values
Results and discussion
Fig 1 b and c are not clearly visible
In figure 3 it is reported the score plot showing the distribution of the samples on the first two principal components; in my opinion it could also be interesting to show the loading plot and comment the positioning of the variables and their relationship with the samples.
E-nose analysis: it might be interesting to process the electronic nose data by PCA together with those collected by GC-MS analysis in order to identify the relationship between the nose sensors and the volatile compounds identified by GC
E-tongue analysis: in order to understand the levels of similarity or difference between the tastes of the analyzed samples, e-tongue data should be analyzed by PCA; moreover, for the characterization of the flavor a final data processing should be made considering together the e-nose and e-tongue data.
Author Response
Thanks for reviewers’ comments concerning our manuscript entitled “E-nose, E-tongue combined with GC-IMS to analyze the influence of key additives during processing on the flavor of infant formula” (ID: foods-1988821). It is clear that you have a deep knowledge of the research in the field and ask very specialized questions. Those comments are all valuable and very helpful for revising and improving our paper, as well as the important guiding significance to our research. The responses to the reviewer’s comments have been highlighted in yellow throughout the manuscript. We hope the changes have addressed all the shortcomings outlined. Below you could find our point-by-point response to the reviewers’ comments.
- line 13-14:e-tongue does not reveal volatile compounds… please change the sentence
Author response: Thanks for the suggestions. We have changed “identify the volatile substances” to “evaluate the flavor”. Please check line 14 in the revised manuscript.
- line 20:what is the meaning of “richness of aftertaste”? have never heard this definition in relation to e-tongue sensory attributes.
Author response: Thanks for the suggestions. AAE sensor used to evaluate the umami sensory attribute, and the aftertaste of umami was described as richness. Umami substances like MSG or peptide have a slight aftertaste, sometimes called “richness”. This may be due to their slight hydrophobicity, helping adsorption on the tongue and causing a lasting slight aftertaste. The definition of “richness” has been mentioned in mangy literatures, such as Laureati at al., 2010, Wang et al., 2022, and Li et al., 2023.
Laureati, M., Buratti, S., Bassoli, A., Borgonovo, G., Pagliarini, E. Discrimination and characterisation of three cultivars of Perilla frutescens by means of sensory descriptors and electronic nose and tongue analysis. Food Research International, 2010, 43: 959-964. https://doi.org/10.1016/j.foodres.2010.01.024.
Li, H.Y., Wang, Y., Zhang, J.X., Li, X.P., Wang, J.X., Yi, S.N., Zhu, W. H., Xu, Y. X., Li, J. R. Prediction of the freshness of horse mackerel (Trachurus japonicus) using E-nose, E-tongue, and colorimeter based on biochemical indexes analyzed during frozen storage of whole fish. Food Chemistry, 2023, 402: 134325. https://doi.org/10.1016/j.foodchem.2022.134325.
Wang, B.Y., Qu, F. F., Wang, P. Q., Zhao, L., Wang, Z., Han, Y. H., Zhang, X. F. Characterization analysis of flavor compounds in green teas at different drying temperature. LWT-Food Science and Technology, 2022, 161: 113394. https://doi.org/10.1016/j.lwt.2022.113394.
- Line 31-33: The reference 2 is about off-flavour in meat and meat products, please look for a more suitable reference.
Author response: Thanks for the suggestions. We have changed this reference. Please check ref. 2 in the revised manuscript.
- Line 61:different substrates ?????? explain better
Author response: Thanks for the suggestions. After explored literature we found that infant formula is a complex matrix usually based on the hydrolysate of milk, goat milk, soymilk or whey proteins and supplemented with appropriate compounds. We have changed “different substrates” to “different breeds or raw materials”. Please check line 61 in the revised manuscript.
- Line 64-66:“In this study, we obtained all the added ingredients in the production of infant formula, which combined with intelligent sensory technology, the sensory quality differences between key additives in the milk powder production process”. It is not clear the meaning of this sentence, English needs to be revised.
Author response: Thanks for the suggestions. We have changed this sentence. “In this study, we obtained all the added ingredients in the production of infant formula stage 1 and discussed the sensory quality differences between key additives through GC-IMS combined with intelligent sensory technology(electronic nose and electronic tongue). ” Please check lines 69-72 in the revised manuscript.
- Line 77: “whey protein powder hydrolyzed powder (HWP)”-do you mean “hydrolyzed whey protein powder”?
Author response: Thanks for the suggestions. Yes, that's exactly what I want to say. We have changed all the words “whey protein powder hydrolyzed powder” to “hydrolyzed whey protein powder” in the revised manuscript.
- Line 87:delete “first”
Author response: Thanks for the suggestions. We have deleted “first”. Please check line 92 in the revised manuscript.
- Line 88:“Finally, the flow rate was 150 mL/min for 30 min and held for 5 min”… 30 minutes or 5 minutes? it is not clear!
Author response: Thanks for the suggestions. We have checked and proofread the description of this method. “The column temperature was kept at 60 °C. Nitrogen (99.999%) was used as carrier gas and its flow rate was first set at 2 mL/min for 2 min, then increased to 10 mL/min within 8 min, then increased to 100 mL/min within 10 min, and then increased to 150 mL/min within 10 min, and held for 5 min.” Please check lines 91-94 in the revised manuscript.
- Line 89: “The entire analysis time kept 35 min”- are you sure? According to my calculation the time is: 2+10+20+ (5 or 30) minutes……
Author response: Thanks for the suggestions. We have checked and proofread the description of this method. Please check lines 91-94 in the revised manuscript.
- Line 101: “equilibration was carried out at a speed of 30 r/min for 5 min”. r???? Please explain the procedure better as it is not clear
Author response: Thanks for the suggestions. We have checked and proofread the description of this method. “The 8.0 mL sample was placed into a 20 mL headspace vital. Incubation temperature was 40°C ± 2°C, and equilibration was carried out at a speed of 960 rpm/min for 300 seconds. E–nose was applied with a detection time of 200 s, cleaning time of 300 s, injection flow rate of 300 mL/min. After per sample analysis, the system was purged with filtered air for 300 seconds before the next sample injection to reestablish the instrument baseline. To ensure the accuracy of the E-nose test results, three groups were performed in parallel for each sample.” Please check lines 102-109 in the revised manuscript.
- Line 106:The sampling time was 200 seconds and the sensor signals were collected only after 6 seconds? Normally after 6 seconds of sampling the sensor signals continue to grow and neither the maximum value nor the plateau have been reached.
Author response: Thanks for the suggestions. We collected 120s of data to observe the response of the sample. However, we chose the 6s with stable response curve for further principal component analysis, that is, the last 6s (115~120s) of sample collection. We have changed “the data after 6s of stability” to “stable data”. Please check line 108 in the revised manuscript.
- Line 111: “salty” is written two times and “umami” is not reported
Author response: Thanks for the suggestions. We are very sorry for our negligence. We have changed “salty” to “umami”. Please check line 113 in the revised manuscript.
- Line 121: Please define or introduce a reference that explains how the electric signals were converted into taste values
Author response: Thanks for the suggestions. Detailed information about the association between electronic signals and taste values has been added. “After filtering and correcting the data, E-tongues turn electrical signals into relish signals to reflect the taste information of sample, through a built-in plug-ins. It calculated the theoretical charge density at the membrane surface using Gouy–Chapman theory and Poisson–Boltzmann equation and then investigated the lipid/polymer membrane’s responses to sour, umami, salty, bitter, astringent, rich-ness, bitter aftertaste, and astringent aftertaste.” Please check lines 121-125 in the revised manuscript.
- Fig 1 b and c are not clearly visible
Author response: Thanks for the suggestions. We have changed the figure with better resolution.
- In figure 3it is reported the score plot showing the distribution of the samples on the first two principal components; in my opinion it could also be interesting to show the loading plot and comment the positioning of the variables and their relationship with the samples.
Author response: Thanks for the suggestions. We have changed the score plot to the PCA biplot, and combined score plot and loading plot to explain the role of volatile compounds in distinguishing samples. Please see Figure 3 and lines 240-245.
- E-nose analysis:it might be interesting to process the electronic nose data by PCA together with those collected by GC-MS analysis in order to identify the relationship between the nose sensors and the volatile compounds identified by GC
Author response: Thanks for the suggestions. As reviewer suggested, the PCA loading plot will help us to have a better known the relationship between sensors of electronic nose and volatile compounds detected by GC-IMS. Thus I have made changes as requested by reviewers. The analysis of PCA loading plot about E-nose and volatile compounds has added in the article at lines 274-285.
- E-tongue analysis:in order to understand the levels of similarity or difference between the tastes of the analyzed samples, e-tongue data should be analyzed by PCA; moreover, for the characterization of the flavor a final data processing should be made considering together the e-nose and e-tongue data.
Author response: Thanks for the suggestions. We have added the correlation analysis between E-nose and E-tongue in Figure 8. Please see Figure 8 and lines 334-344.
We appreciate for Editors/Reviewers’ warm work earnestly, and hope that the correction will meet with approval.
Once again, thank you very much for your comments and suggestions
Best regards for you.
Sincerely yours,
Authors

Reviewer 2 Report
The manuscript “E-nose, E-tongue combined with GC-IMS to analyze the influence of key additives during processing on the flavor of infant formula”. The study is good and the findings are satisfactory. Though several points need to address before the acceptance of the MS.
- Keywords should have been opted more wisely. The words that appear in the title such as “infant formula; GC-IMS; flavor; E-tongue; E-nose” should be avoided in the keywords.
- Remove the word triplicate from the line “All the samples were stored in …………… until testing” (lines 78-79).
- Column temperature 60oC (line no 86), is it an isothermal condition or its gradient? Usually, GC operates at a much higher temperature. Check this carefully.
- Also, check the flow rate (line 87) is it 100 ml/ min? It looks weird, check it out carefully.
- What does the author mean by 300 r/min (line 101) is it rpm as mentioned in line 84? If yes then exchange r/min to rpm.
- Lines (100-103) “The 8.0 mL sample …………flow rate of 300 mL/min.” is not clear. Rewrite the lines for clarity and better understanding.
- What is 6s line no? 106????/
- The resolution of Figures 1 and 2 is inferior and hard to read anything. Replace these images with images of high resolution.
- In table 3 write the full of abbreviations such as “MW, RI, RT, and Dt” as a footnote.
- Rectify the typo mistake of Amyl alcohol (line 199).
- Write the full name of the abbreviations such as PCA and FSA while appearing first in the manuscript.
- For the PCA (Figure 3) add the lines of zero coordinates for the x and y axis so that easy to understand the positive and negative correlation between the groups.
- What is the unit for numerical values mentioned in figure 5?
- In table 4, Is there any unit for sourness Saltiness Bitterness Astringency, Umami, etc. if yes add to the table otherwise mentioned as an Arbitrary unit (AU).
- What does the meaning of the alphabet after the numeric values mentioned in table 4? Is it a statistical difference, if so put a footnote of the statement with a test of significance carried out in the column or row with the level of significance.
- Figure 6 is blur draw it. Also add the values of the x and y axis in the respective figures (a, b, and c).
Author Response
Thanks for reviewers’ comments concerning our manuscript entitled “E-nose, E-tongue combined with GC-IMS to analyze the influence of key additives during processing on the flavor of infant formula” (ID: foods-1988821). It is clear that you have a deep knowledge of the research in the field and ask very specialized questions. Those comments are all valuable and very helpful for revising and improving our paper, as well as the important guiding significance to our research. The responses to the reviewer’s comments have been highlighted in yellow throughout the manuscript. We hope the changes have addressed all the shortcomings outlined. Below you could find our point-by-point response to the reviewers’ comments.
- Keywords should have been opted more wisely. The words that appear in the title such as “infant formula; GC-IMS; flavor; E-tongue; E-nose” should be avoided in the keywords.
Author response: Thanks for the suggestions. We have changed “infant formula; GC-IMS; flavor; E-tongue; E-nose” to “Infant formula; headspace-gas chromatography-ion mobility spectrometry; volatile compounds; E-tongue; E-nose”. Please check lines 28-29 in the revised manuscript.
- Remove the word triplicate from the line “All the samples were stored in ……until testing” (lines 78-79).
Author response: Thanks for the suggestions. We have deleted the words. Please check line 83 in the revised manuscript.
- Column temperature 60℃ (line no 86), is it an isothermal condition or its gradient? Usually, GC operates at a much higher temperature. Check this carefully.
Author response: Thanks for the suggestions. The sample was transferred into the column through high-purity nitrogen (99.99%) and introduced into an ionization chamber after elution at 60 °C (isothermal mode). The sample was later scanned in the drift tube, and the VOCs were identified by comparing their retention index and drift time with the standards in the GC-IMS library.
- Also, check the flow rate (line 87) is it 100 ml/ min? It looks weird, check it out carefully.
Author response: Thanks for the suggestions. The flow rate is 100ml/min. We have checked and proofread the description of this method. “The column temperature was kept at 60 °C. Nitrogen (99.999%) was used as carrier gas and its flow rate was first set at 2 mL/min for 2 min, then increased to 10 mL/min within 8 min, then increased to 100 mL/min within 10 min, and then increased to 150 mL/min within 10 min, and held for 5 min.” Please check lines 91-94 in the revised manuscript.
- What does the author mean by 300 r/min (line 101) is it rpm as mentioned in line 84? If yes then exchange r/min to rpm.
Author response: Thanks for the suggestions. We have checked and proofread the description of this method. “The 8.0 mL sample was placed into a 20 mL headspace vital. Incubation temperature was 40°C ± 2°C, and equilibration was carried out at a speed of 960 rpm/min for 300 seconds. E–nose was applied with a detection time of 200 s, cleaning time of 300 s, injection flow rate of 300 mL/min. After per sample analysis, the system was purged with filtered air for 300 seconds before the next sample injection to reestablish the instrument baseline. To ensure the accuracy of the E-nose test results, three groups were performed in parallel for each sample, and stable data was selected for statistical analysis during the measurement process.” Please check lines 102-109 in the revised manuscript.
- Lines (100-103) “The 8.0 mL sample …………flow rate of 300 mL/min.” is not clear. Rewrite the lines for clarity and better understanding.
Author response: Thanks for the suggestions. We have checked and proofread the description of this method. Please check lines 102-109 in the revised manuscript.
- What is 6s line no? 106????/
Author response: Thanks for the suggestions. We collected 120s of data to observe the response of the sample. However, we chose the 6s with stable response curve for further principal component analysis, that is, the last 6s (115~120s) of sample collection. We have changed “the data after 6s of stability” to “stable data”. Please check line 108 in the revised manuscript.
- The resolution of Figures 1 and 2 is inferior and hard to read anything. Replace these images with images of high resolution.
Author response: Thanks for the suggestions. We have changed the figures with better resolution.
- In table 3 write the full of abbreviations such as “MW, RI, RT, and Dt” as a footnote.
Author response: Thanks for the suggestions. We are very sorry for our negligence. We have added the full of abbreviations as a footnote. Please check line 177 in the revised manuscript.
- Rectify the typo mistake of Amyl alcohol (line 199).
Author response: Thanks for the suggestions. We are very sorry for our negligence. We have changed this word. Please check line 204 in the revised manuscript.
- Write the full name of the abbreviations such as PCA and FSA while appearing first in the manuscript.
Author response: Thanks for the suggestions. We have added the full name of the abbreviations. Please check line 245 in the revised manuscript.
- For the PCA (Figure 3) add the lines of zero coordinates for the x and y axis so that easy to understand the positive and negative correlation between the groups.
Author response: Thanks for the suggestions. We have modified Figure 3.
- What is the unit for numerical values mentioned in figure 5?
Author response: Thanks for the suggestions. There is no unit for numerical values mentioned in Figure 5. It represents the response value of the electronic nose sensors.
- In table 4, Is there any unit for sourness Saltiness Bitterness Astringency, Umami, etc. if yes add to the table otherwise mentioned as an Arbitrary unit (AU).
Author response: Thanks for the suggestions. Thanks for the suggestions. There are no unit for these attributes. As Reviewer suggested, we added arbitrary unit in Table 4.
- What does the meaning of the alphabet after the numeric values mentioned in table 4? Is it a statistical difference, if so put a footnote of the statement with a test of significance carried out in the column or row with the level of significance.
Author response: Thanks for the suggestions. We are very sorry for our negligence. We have added the footnote. "Note: Different lowercase letters (a–d) in the same row indicate significant differences between different taste values (p < 0.05);". Please check lines 316-317 in the revised manuscript.
- Figure 6 is blur draw it. Also add the values of the x and y axis in the respective figures (a, b, and c).
Author response: Thanks for the suggestions. We have changed the figure with better resolution. The x-axis and y-axis are added to the scatter plot, and the principal component values are added to the PCA plot. Please check Figure 6.
We appreciate for Editors/Reviewers’ warm work earnestly, and hope that the correction will meet with approval.
Once again, thank you very much for your comments and suggestions.
Best regards for you.
Sincerely yours,
Authors

Reviewer 3 Report
The article presents an analysis of the compounds in milk powder after different processing stages.
It is a basic but interesting result, however, there is no novelty in these results.
To improve the article, I suggest discussing the mechanisms by which the samples may change in composition and flavors.
Author Response
Thanks for reviewers’ comments concerning our manuscript entitled “E-nose, E-tongue combined with GC-IMS to analyze the influence of key additives during processing on the flavor of infant formula” (ID: foods-1988821). It is clear that you have a deep knowledge of the research in the field and ask very specialized questions. Those comments are all valuable and very helpful for revising and improving our paper, as well as the important guiding significance to our research. The responses to the reviewer’s comments have been highlighted in yellow throughout the manuscript. We hope the changes have addressed all the shortcomings outlined. Below you could find our point-by-point response to the reviewers’ comments.
- The article presents an analysis of the compounds in milk powder after different processing stages. It is a basic but interesting result, however, there is no novelty in these results. To improve the article, I suggest discussing the mechanisms by which the samples may change in composition and flavors.
Author response: Thanks for the suggestions. We have added some contents in the foreword about the reason why we chose the first stage of infant formula as the research object and introduced the background of the first stage infant formula as the most basic complementary food. Infant formula stage 1 used to supplement nutrition for newborn babies aged 0-6 months. Compared with the Infant formula in other stages, the added supplements are the least, so the aroma components are simpler, relatively. We selected the most basic products in infant formula for research and discussion. We hope to get some inspiration from it. By analyzing the basic products, we can not only better understand the importance of raw materials, but also provide basic data support for subsequent research. According to your suggestion, we have added some analysis on mechanism in the analysis and discussion section.
We appreciate for Editors/Reviewers’ warm work earnestly, and hope that the correction will meet with approval.
Once again, thank you very much for your comments and suggestions.
Best regards for you.
Sincerely yours,
Authors

Reviewer 4 Report
A very interesting and important study to analyze the effect of key additives during processing on the taste of babies
formula. This study revealed the qualities and flavor differences of key additives in the production of powdered infant formula.
The study provides a great deal of valuable information on the composition of the different types of milk used in the production of infant formula. The only thing I am missing is why the authors chose only one type of end product for the study or did not describe it. In China, are all infant formulas (type 1) identical in composition and derived from the same starting products? perhaps in the future it would be necessary to examine the different infants formulas produced from different starting products and compare their composition. This is a suggestion for the future, which can also be treated as a limitation of the presented work.
There is no description of the strenght and limitation of the presented studies.
Author Response
Thanks for reviewers’ comments concerning our manuscript entitled “E-nose, E-tongue combined with GC-IMS to analyze the influence of key additives during processing on the flavor of infant formula” (ID: foods-1988821). It is clear that you have a deep knowledge of the research in the field and ask very specialized questions. Those comments are all valuable and very helpful for revising and improving our paper, as well as the important guiding significance to our research. The responses to the reviewer’s comments have been highlighted in yellow throughout the manuscript. We hope the changes have addressed all the shortcomings outlined. Below you could find our point-by-point response to the reviewers’ comments.
- The study provides a great deal of valuable information on the composition of the different types of milk used in the production of infant formula. The only thing I am missing is why the authors chose only one type of end product for the study or did not describe it. In China, are all infant formulas (type 1) identical in composition and derived from the same starting products? perhaps in the future it would be necessary to examine the different infant formulas produced from different starting products and compare their composition. This is a suggestion for the future, which can also be treated as a limitation of the presented work.
Author response: Thanks for the suggestions. We have added some contents in the foreword about the reason why we chose the first stage of infant formula as the research object and introduced the background of the first stage infant formula as the most basic complementary food. Infant formula stage 1 used to supplement nutrition for newborn babies aged 0-6 months. Compared with the Infant formula in other stages, the added supplements are the least, so the aroma components are simpler, relatively. We selected the most basic products in infant formula for research and discussion. We hope to get some inspiration from it. By analyzing the basic products, we can not only better understand the importance of raw materials, but also pro-vide basic data support for subsequent research. In the next work, we will try our best to analyze more flavors of milk powder at different stages, and comprehensively evaluate the quality of complementary foods for infants and young children, speculate the factors that may bear on the sensory quality of infant formula milk powder products, and provide references for flavor quality control in the production process.
We appreciate for Editors/Reviewers’ warm work earnestly, and hope that the correction will meet with approval.
Once again, thank you very much for your comments and suggestions.
Best regards for you.
Sincerely yours,
Authors

Round 2
Reviewer 1 Report
- Please check the main group of W3C e-nose sensor in table 1: ammonia-C6H6??????
- Table 1 first row: I don't understand the meaning of "sensitive to long chain alkanes" as a title
-Figure 6 is not visible, it must be enlarged
Author Response
Thanks for reviewers’ comments concerning our manuscript entitled “E-nose, E-tongue combined with GC-IMS to analyze the influence of key additives during processing on the flavor of infant formula” (ID: foods-1988821). The responses to the reviewer’s comments have been highlighted in blue throughout the manuscript. We hope the changes have addressed all the shortcomings outlined. Below you could find our point-by-point response to the reviewers’ comments.
- Please check the main group of W3C e-nose sensor in table 1: ammonia-C6H6??????
Table 1 first row: I don't understand the meaning of "sensitive to long chain alkanes" as a titleline
Author response: Thanks for the suggestions. We are very sorry for our negligence. We have changed “sensitive to long chain alkanes” to “Performance Description (Sensitivity to)”. Please check table 1 in the revised manuscript.
- Figure 6 is not visible, it must be enlarged
Author response: Thanks for the suggestions. We have changed the figure 6 with better resolution, and we have uploaded the original image of Figure 6 as an attachment.
Reviewer 2 Report
The asked changes has been done, though the resolution of figure 6 in low. I recommend for the publication of the article after improvement of the figure 6 resolution.
Author Response
Thanks for reviewers’ comments concerning our manuscript entitled “E-nose, E-tongue combined with GC-IMS to analyze the influence of key additives during processing on the flavor of infant formula” (ID: foods-1988821). The responses to the reviewer’s comments have been highlighted in blue throughout the manuscript. We hope the changes have addressed all the shortcomings outlined. Below you could find our point-by-point response to the reviewers’ comments.
- The asked changes has been done, though the resolution of figure 6 in low. I recommend for the publication of the article after improvement of the figure 6 resolution.
Author response: Thanks for the suggestions. We have changed the figure 6 with better resolution.